# Enhanced Formaldehyde Removal from Air Using Fully Biodegradable Chitosan Grafted β-Cyclodextrin Adsorbent with Weak Chemical Interaction

**DOI:** 10.3390/polym11020276

**Published:** 2019-02-06

**Authors:** Zujin Yang, Hongchen Miao, Zebao Rui, Hongbing Ji

**Affiliations:** 1School of Chemical Engineering and Technology, Sun Yat-sen University, Zhuhai 519082, China; yangzj3@mail.sysu.edu.cn; 2Huizhou Research Institute of Sun Yat-sen University, Huizhou 516216, China; 3Fine Chemical Industry Research Institute, The Key Laboratory of Low-carbon Chemistry & Energy Conservation of Guangdong Province, School of Chemistry, Sun Yat-sen University, Guangzhou 510275, China; miaohongcheng@126.com; 4School of Chemical Engineering, Guangdong University of Petrochemical Technology, Maoming 525000, China

**Keywords:** formaldehyde adsorption, CGC, synergistic effects, weak chemical interaction, β-cyclodextrin

## Abstract

Formaldehyde (HCHO) is an important indoor air pollutant. Herein, a fully biodegradable adsorbent was synthesized by the crosslinking reaction of β-cyclodextrin (β-CD) and chitosan via glutaraldehyde (CGC). The as-prepared CGC showed large adsorption capacities for gaseous formaldehyde. To clarify the adsorption performance of the as-synthesized HCHO adsorbents, changing the adsorption parameters performed various continuous flow adsorption tests. It was found that the adsorption data agreed best with the Freundlich isotherm, and the HCHO adsorption kinetic data fitted well with the pseudo second order model. The breakthrough curves indicated that the HCHO adsorbing capacity of CGC was up to 15.5 mg/g, with the inlet HCHO concentration of 46.1 mg/m^3^, GHSV of 28 mL/min, and temperature of 20 °C. The regeneration and reusability of the adsorbent were evaluated and CGC was found to retain its adsorptive capacity after four cycles. The introduction of β-CD was a key factor for the satisfied HCHO adsorption performance of CGC. A plausible HCHO adsorption mechanism by CGC with the consideration of the synergistic effects of Schiff base reaction and the hydrogen bonding interaction was proposed based on in situ DRIFTS studies. The present study suggests that CGC is a promising adsorbent for the indoor formaldehyde treatment.

## 1. Introduction

In recent years, indoor air pollution has attracted more and more attentions. Volatile organic compounds (VOCs) are the main source of indoor air pollution, and formaldehyde (HCHO) has been considered as one of the most widespread VOCs, which are often released from building and furnishing materials [1,2,3]. HCHO is a kind of hazardous gas that is dangerous to human health, and long-term exposure to HCHO can cause a variety of health effects [4,5]. Thus, it is necessary to develop a practical and effective technique to eliminate indoor HCHO.

Various technologies have been developed for scavenging indoor HCHO pollutant, including absorption, decomposition, photocatalysis, and catalytic oxidation [6,7,8]. Among these methods, adsorption and catalytic oxidation are widely accepted as effective methods for indoor HCHO eliminating. Although catalytic oxidation can completely oxidize HCHO into CO_2_ and H_2_O, their practical applications have been restricted due to the high cost of precious metals [9,10]. Adsorption is another promising method, because of its low cost and ease of operation. Activated carbon is one of the most common materials that are extensively used as an adsorbent to clean air contaminants due to its suitable textural properties, such as high specific surface area and microporous volume [11,12]. However, the poor adsorption selectivity, recyclability, and efficiency restrict the application of activated carbon. Recently, many efforts have been made to develop new classes of chemically functionalized, low-cost materials for HCHO adsorption [1,13]. When considering that HCHO molecules hold the polarity and reactivity with base, many researches focused on surface modification of adsorbents with organic amines [13,14]. For example, some functional groups are known to promote HCHO removal efficiency through the Schiff base reaction between amines and aldehydes.

Chitosan is a modified biopolymer that was obtained from chitin, which is one of the most abundant natural amino polysaccharide, by deacetylation, and is composed primarily of repeating β-(1,4)-2-amino-2-deoxy-d-glucose (or d-glucosamine) units. The advantages of chitosan include low cost, ease of polymerization and functionalization, as well as good stability [13,14,15]. Recent studies showed that chitosan-supported adsorbents were effective materials for the removal of HCHO from indoor air. Nuasaen et al. [16] developed hollow latex particles that were functionalized with chitosan for the adsorption and removal of formaldehyde from indoor air. Yang et al. [17] prepared chitosan/silver nano-composites for the removal of formaldehyde in air. However, as an adsorbent, chitosan also has some limitations, such as poor chelating functionality, small surface area, and low pore volume [18,19]. Therefore, new preparation methods or a design strategy are required to increase the surface area and internal pore volumes of chitosan as well as introduce functional groups onto its surface to enhance its HCHO removal efficiency. 

β-Cyclodextrin (abbreviated as β-CD), composed of seven α-1,4-linked-glucopyranose units, is a torus-shaped cyclic oligosaccharide with a hydrophilic exterior and a hydrophobic internal cavity. It can form inclusion complexes with organic and inorganic molecules through host-guest interactions [20,21]. The combination of the hydrophobic cavity and hydrophilic exterior of β-CD and electrostatic attraction of chitosan are usually used to improve the adsorption capacity of adsorbents. In addition, the stability and recoverability of adsorbents may also be improved after grafting β-CD onto the chitosan (CGC) [22,23]. 

In this study, we prepared an insoluble CGC bead by the cross-linking reaction between β-CD and chitosan in acidic aqueous solution with glutaraldehyde as the cross-linker. It was used as an adsorbent for the removal of indoor HCHO. The structure of the adsorbent was confirmed by various physico-chemical techniques. The adsorption/desorption of HCHO on CGC were investigated in the continuous mode of operation. The stability, adsorption kinetics, and adsorption isotherm of CGC were also evaluated. The adsorption mechanism was proposed based on the hydrogen bonding interaction between the active hydroxyl groups of β-CD and HCHO, as well as the electrostatic interaction with HCHO by chitosan units. 

## 2. Materials and Methods 

### 2.1. Preparation of CGC

CGC was prepared according to previous method with minor modifications [24]. A typical procedure was described as: 4.0 g of chitosan was stirred in 200 mL of 0.1 mol/L HCl solution for 1 h and then mixed with 200 mL of β-CD aqueous solution containing 24 g of β-CD. The mixture was stirred for 1 h at 25 °C, followed by the dropwise addition of 25 wt % glutaraldehyde solution (~24 mL). The solution was then magnetically stirred at 70 °C for 6 h, followed by adding 70 mL of 1 mol/L NaOH aqueous solution into the reaction solution to precipitate the product. The precipitate was then thoroughly rinsed with distilled water and acetone, dried in vacuum at 60 °C to a constant weight, and then kept in a desiccator for the following applications. The amount of β-CD that was grafted on the surface of chitosan was calculated according to the previously reported method [25].

### 2.2. Characterization of CGC

The particle size distribution of the adsorbent was analyzed by a Mastersizer 2000 laser particle size analyzer (Malvern, Worcestershire, UK). FTIR spectroscopic measurements were performed in the range 4000–400 cm^−1^ on a Bruker Tensor 37 FTIR Spectrometer (Bruker, Karlsruhe, German). TG/DTA experiments were performed on a Netzsch STA-449C thermal analysis system (Bavaria, German) by heating to 800 °C at the rate of 10 °C/min under constant N_2_ flow. Prior to experiments, the TG-DTA apparatus was calibrated and then purged with nitrogen for 30 min before each run. SEM was used to analyze the surface morphology of the adsorbent on Hitachi S–520 (Hitachi, Tokyo, Japan) with an accelerating voltage of 15 kV. The adsorbent was coated with a thin layer of sputtered gold prior to examination. The X-ray power diffraction (XRD) patterns were measured on a Rigaku D/MAX 2200 instrument (Rigaku, Tokyo, Japan) with Cu *Kα* radiation at a scanning speed of 4°/min and a step size of 0.02°. BET surface areas of the adsorbents were measured by N_2_ adsorption tests at 77 K on ASAP 2020 equipment (Micromeritics, Norcross, GA, USA). Before N_2_ adsorption tests, the samples were degassed in vacuum at 300 °C for 2 h. In-situ DRIFTS (or diffuse reflectance infrared Fourier transform) studies were operated on a Bruker EQVINOX-55 FFT spectroscope (Bruker, Karlsruhe, German) apparatus with a MCT detector. About 10 mg of finely ground adsorbent was held in the ceramic crucible and then placed in the in situ chamber. The inlet gas was 100 mL/min of N_2_ containing ~50 ppm HCHO. The spectra were recorded under reaction conditions after 64 scans with a resolution of 4 cm^−1^. 

### 2.3. Adsorption Performance Test

The physical properties of the adsorbent and typical adsorption operation conditions are summarized in Table 1. The adsorption tests were performed using a shell-tube structured quartz fixed-bed reactor under atmospheric pressure. The schematic diagram of the adsorption/desorption experiments is shown in Scheme 1. The adsorbent (~0.5 g) was packed in the tube side. The temperature was controlled by the recycling water in the shell side. Air from a cylinder was separated into two lines. One line flowed into a saturator that was filled with HCHO solution (~35% HCHO) and was then mixed with the volatilized gas and the gas from another line. The concentration of HCHO was adjusted by controlling the temperature of saturator and the air flow rate in the two lines. During the HCHO desorption experiments or the regeneration of the adsorbent, the air flow line through the HCHO solution was closed. The HCHO concentration in the inlet and outlet gas streams was analyzed with the phenol spectrophotometric method [26,27]. The amount of adsorbed HCHO was calculated based on the concentration change of HCHO between the influent and effluent gas streams.

The adsorption capacity *q_e_* was calculated by the following equation,
(1)$$q_{e}=\int_{0}^{t}\frac{q_{v}\left(C_{inlet}-C_{outlet}\right)}{w}dt$$
where *q_e_* is HCHO adsorption capacity (mg/g), *q_v_* is the inlet gas volumetric flow rate during adsorption (mL/min), *C_inlet_* and *C_outlet_* are the influent and the effluent concentration of HCHO at a certain time (mg/m^3^), *t* is the adsorption time (min), and *w* is the weight of virgin adsorbent (g).

## 3. Results and Discussion

### 3.1. Characterization of CGC

#### 3.1.1. FTIR Spectral Analysis

Grafting of β-CD onto chitosan was investigated by FTIR analysis, as presented in Figure 1. The FTIR spectrum of β-CD shows the characteristic absorption bands at 3390 cm^−1^ (–OH stretching vibration), 2935 cm^−1^ (–CH_2_ stretching vibration), 1648 cm^−1^ (C–C vibrations of polysaccharides), 1160 cm^−1^ (C–O stretching vibration), and 1030 cm^−1^ (C–O–C stretching vibration) [28]. The IR spectrum of chitosan indicates the –OH stretching broad band at 3351 cm^−1^, which overlaps the –NH stretching in the same region. The peak at 2864 cm^−1^ is mainly attributed to the –CH and –CH_2_ stretching vibrations. The broad bands of 1655 cm^−1^ and 1578 cm^−1^ are related to the C–O stretching vibration of –NHCO– (amide I) and the N–H bending of–NH_2_ (amide II), respectively [29]. When β-CD was grafted onto chitosan, the relative intensity of the band at 3400 cm^−1^ of CGC at 3351 cm^−1^ is much broader than that of chitosan, which is attributed to the presence of more –OH groups in CGC that result from the coupling of β-CD. The –CH_2_ stretching vibration band at 2935 cm^−1^ and the C–H stretching peak at 2864 cm^−1^ were observed on CGC. In comparison with chitosan, a sharp peak at 1650 cm^−1^ was observed in CGC and the band at 1598 cm^−1^ decreases significantly due to the C=N vibrations from Schiff base imines. In addition, the characteristic peak at 1648 cm^−1^ due to the C–C vibrations of polysaccharides of β-CD appears in CGC, implying that β-CD is successfully grafted onto the surface of chitosan. Similar phenomena were observed in the literature [24]. 

#### 3.1.2. TG/DTA

TG/DTA curves of CGC are shown in Figure 2. CGC exhibits three mass loss steps. The first mass loss step at around 75 °C is due to the elimination of physically absorbed water. The second stage that started at about 250 °C with a mass loss of around 40% is related to the heat decomposition of CGC, and the third stage that is above 400 °C may be attributed to the carbonization process. The results indicate that CGC is thermally stable up to 250 °C.

#### 3.1.3. SEM

SEM images for CGC at different magnifications are presented in Figure 3. The low-magnification SEM image in Figure 3a clearly shows the typical lamellar structure with lots of pores being embedded of the as-prepared CGC. In Figure 3b, some dense microstructure with some porosity was observed. The BET surface area of the as-prepared CGC was measured to be 10.8 m^2^/g, higher than pure chitosan (2.7 m^2^/g). The pore diameter of CGC is 7.81 nm, which is almost twice that of pure chitosan (3.83 nm). The pore volume of CGC also increases from 0.0089 for pure chitosan to 0.02 cm^3^/g. Figure 3a also indicates that β-CD has a dense crystal structure, and after immobilization onto chitosan, CGC has an ordered porous structure, with a pore size range of 5–20 nm, indicating that it is beneficial for the diffusion and adsorption of the pollutant gas.

#### 3.1.4. Powder XRD Pattern

X-ray power diffraction (XRD) patterns of the samples were shown in Figure 4. β-CD in its crystalline form displays diffraction peaks at 2*θ* values of 4.9°, 10.7°, 12.9°, 19.8°, 20.9°, 22.8°, 24.3°, and 36.0°. On the other hand, the XRD pattern of chitosan shows characteristic peaks at 2*θ* = 10.0° and 21.3°, which correspond to the hydrated crystalline structure and the existence of an amorphous structure [30]. The 2*θ* peak at 10.0° disappears, and the characteristic peak at 2*θ* = 21.3° decreases and it obviously gets broadened for CGC, indicating the amorphous state of the as-synthesized polymer material. The results indicate that chemical modification of chitosan and the introduction of β-CD onto chitosan backbone destroyed its original crystallinity to some extent.

### 3.2. Equilibrium Adsorption Studies of HCHO

#### 3.2.1. Adsorption Isotherms

Figure 5 shows the adsorption isotherms of HCHO on CGC at different temperatures. As shown, the adsorption capacities of HCHO on CGC significantly decrease with increasing temperature, indicating that HCHO adsorption on CGC is an exothermic process. The Langmuir and Freundlich adsorption models were used to fit the equilibrium HCHO adsorption data of CGC to at concentrations that range from 5 to 45 mg/m^3^ [31].

The Langmuir model is usually used to calculate the monolayer adsorption capacity of the adsorbent, which is expressed as,
(2)$$\frac{C_{e}}{q_{e}}=\frac{C_{e}}{q_{m}}+\frac{1}{K_{L}q_{m}}$$
where *q_e_* (mg/g) is the equilibrium amount of adsorbed HCHO by CGC, *C_e_* is equilibrium HCHO concentration (mg/m^3^) in air, *q_m_* is the maximum adsorption capacity (mg/g), and *K_L_* is the Langmuir constant for monolayer adsorption (m^3^/mg). *q_m_* and *K_L_* are calculated from the intercept and slope of linear plot of *C_e_*/*q_e_* versus *C_e_*.

The Freundlich model was adopted for evaluating the multilayer adsorption on a heterogeneous adsorbent surface, which can be expressed as
(3)lnqe=lnKf+1nlnCe
where *K_f_* and *n* are the Freundlich isotherm constant and adsorption intensity, respectively. They can be regressed from the linear plot of ln *q_e_* against ln *C_e_*, respectively.

To further understand the adsorption mechanism, the D–R isotherm model was also applied for the adsorption process. It can be determined by
(4)$$\ln q_{e}=\ln q_{m}-\beta \varepsilon^{2}$$

Here, *β* is a constant that is related to the mean free energy of adsorption (mol^2^/kJ^2^) and *ε* is the Polanyi potential, which can be calculated by
(5)ε=RTln(1+1Ce)
where *R* is the gas constant and *T* is the absolute temperature. Adsorption capacity *q_m_* (mg/g) and *β* can be obtained from the slope of the plot of ln *q_e_* versus *ε*^2^. The mean free energy of adsorption (*E*) can be calculated from the *β* value with the following equation,
(6)$$E=\frac{1}{{\left(2\beta \right)}^{0.5}}$$

The parameters calculated from the Langmuir and Freundlich models, as well as D–R models, were listed in Table 2. Table 2 indicates the correlation coefficients (*R*^2^) of the linear form for Freundlich model is much higher than those of the Langmuir and D–R models within the temperature studied. The adsorption isotherm of HCHO on CGC is best described by the Freundlich isotherm. The results of the D–R model show that the adsorption capacity (*q_m_*) decreases with the increase of temperatures from 20 to 60 °C, and the maximum adsorption capacities by the Langmuir model are 15.48, 9.71, and 6.33 mg/g, respectively. A comparison between CGC with those recently reported adsorbents shows that CGC holds significantly higher HCHO adsorption capacity per unit BET area (Table 3) [32,33,34,35].

As listed in Table 2, the calculated parameter *n* in the Freundlich equations are more than 1 for the Freundlich equations at 20 and 40 °C, indicating that CGC can efficiently enhance the adsorption of HCHO at a low temperature. In addition, the results indicate a multi-molecular layer HCHO adsorption on CGC. The D-R isotherm model is also used to analyze the mechanism. It can give some information for the adsorption process being chemical or physical adsorption. The mean free energy of adsorption (*E*) can be employed to classify the adsorption. *E* in the range of 1–8 kJ/mol follows physical adsorption; *E* between 8 to 16 kJ/mol means chemical ion exchange, while *E* in the range of 20–40 kJ/mol is indicative of chemisorptions [36]. As listed in Table 2, the values of *E* are 10.70, 8.51, and 1.99 kJ/mol for 20, 40, and 60 °C, respectively. The results implies that the adsorption of HCHO on CGC is close to a physicochemical adsorption process, but with a weak chemical interaction.

#### 3.2.2. Adsorption Kinetics for HCHO

Figure 6 shows the measured HCHO adsorption as a function of contact time over CGC at 20, 40, and 60 °C. The adsorption is initially rapid and it then became slow with the increase in contact time at various temperatures. It is attributed to the greater amount of external adsorption sites of CGC that are available at the beginning of the adsorption, and the remaining vacant surface sites are difficult to be taken due to the repulsive forces between HCHO molecule and CGC. In addition, it was observed that maximum adsorption capacities took place within first 40, 60, and 105 h at 60, 40, and 20 °C, respectively. These results indicate that the equilibrium time and the HCHO adsorption capacities decrease with increasing temperature. 

Kinetic models are helpful to identify the adsorption mechanism of HCHO adsorption and desorption performance of the adsorbent. In this study, pseudo first-order kinetic, pseudo-second-order kinetic, and intra-particle diffusion models were, respectively, checked to fit the kinetic data and explain the corresponding HCHO adsorption mechanism [37,38]. The best-fitted model was then selected on the basis of the correlation coefficient (*R*^2^) of the linear regression of the experimental data with the proposed models. The pseudo-first-order kinetic model assumes reversible interactions between the gas and solid surfaces. This model can be expressed as:(7)$$\ln(q_{e}-q_{t})=\ln q_{e}-k_{1}t$$
where *q_e_* and *q_t_* are the amounts of HCHO that are adsorbed (mg/g) at equilibrium and time *t* (*h*), respectively, and *k*_1_ is the rate constant for the pseudo-first-order kinetic model (h^−1^). The intercept and the slope of linear plots of ln (*q_e_* − *q_t_*) versus *t* are used to calculate *q_e_* and *k*_1_.

The pseudo-second-order kinetic model is proposed by assuming that chemical interactions control the adsorption kinetics and it is expressed as,
(8)$$\frac{t}{q_{t}}=\frac{1}{k_{2}q_{e}^{2}}+\frac{t}{q_{e}}$$
where *k*_2_ is the rate constant (g/(mg h)) of the pseudo-second-order kinetic model for the adsorption. Furthermore, *q_e_* and *k*_2_ are determined from the slope and the intercept of the linear plot of *t*/*q_t_* against *t*.

An intra-particle diffusion model is also employed to analyze the kinetic data.
(9)$$q_{t}=K_{p}t^{0.5}+C$$
where *K_P_* is the intra-particle diffusion rate constant (mg/g min^1/2^) and *C* is related to the boundary layer thickness (mg/g), which is calculated from the slope of the linear plots of *q_t_* versus *t*^1/2^.

These kinetic models were used to fit the HCHO adsorption data and they are compared in Figure 7, which have been used for the determination of gases mass transfer coefficients on various adsorbents [39,40]. The corresponding parameters and the correlation coefficients (*R*^2^) for the three models are summarized in Table 4. *R*^2^, for the pseudo-second-order adsorption model, were higher (*R*^2^ > 0.99) than those of the pseudo-first-order and intra-particle diffusion models for all of the studied temperatures. In addition, *q_e_* (mg/g), as calculated from the pseudo-second-order kinetic model, agrees better with the experimentally obtained *q_t_* values than those calculated with the pseudo-first-order model and intra-particle diffusion model. Therefore, the pseudo-second-order kinetic model reasonably fits the experimental kinetic curves of HCHO adsorption at all studied temperatures, indicating the weak interactions, such as van der Waals force, hydrogen bonding, as well as hydrophobic interaction, play a crucial role in the adsorption processes and the adsorption capacity is proportional to the active sites of CGC.

Thermodynamic parameters, such as the change in Gibbs free energy (Δ*G*), enthalpy (Δ*H*), and entropy (Δ*S*), can be determined using the equations [41]:(10)ΔG=−RTlnK
(11)$$\ln K=-\frac{\Delta H}{RT}+\frac{\Delta S}{R}$$
where *K* is the Langmuir equilibrium constant; *R* and *T* are the gas constant and the temperature (K), respectively. Δ*H* and Δ*S* can be calculated from the slope and intercept of the van’t Hoff plots of ln (*K*) versus 1/*T*.

Table 5 lists the calculated results. The negative Δ*G* values indicate that the HCHO adsorption over CGC at the temperatures studied is feasible and thermodynamically spontaneous. The Δ*G* values increase with increasing temperature, demonstrating that the adsorption of HCHO is more favorable at a low temperature, which agrees well with the experimental findings listed above. The negative value of Δ*H* shows that the adsorption is an exothermic process. The negative Δ*S* indicates the decrease in randomness at the solid/gas interface during adsorption of HCHO on CGC.

The rate constants (*k*_2_) that were obtained from the kinetic model increase with increasing adsorption temperatures, as shown in Table 4. The dependence between the rate constant and temperature can be described as:(12)$$k=k_{0}\exp(\frac{-E_{a}}{RT})$$
where *k* is the adsorption rate constant of (h^−1^), *k*_0_ is the pre-exponential factor, *E_a_* is the apparent activation energy, *R* is the gas constant, and *T* is the absolute temperature. A good linearity between ln *k* and 1/*T* with coefficients of determination (*R*^2^) larger than 0.98 was obtained for *k*_2_ values that were obtained in this work. The activation energy for the HCHO adsorption process was lower than that of previous data [42], indicating the good HCHO adsorption property of CGC.

Breakthrough experiments were performed for β-CD, chitosan, and CGC at various adsorption temperatures and feed flow rates. Figure 8 presents the effluent HCHO concentration as a function of time (leakage curve) for each adsorbent at 20, 40, and 60 °C, with varying flow rates of the feed gas from 28 to 84 mL/min in the feed gas. To demonstrate the adsorption leakage behavior of HCHO, the following parameters, *t*_b_, *t_e_*, and *L_MTZ_*, were used to analyze the breakthrough curves:(13)$$L_{MTZ}=L\left[\frac{t_{e}-t_{b}}{t_{e}-0.5\left(t_{e}-t_{b}\right)}\right]$$
where *t_b_* refers to the time corresponding to *C*_outlet_/*C*_inlet_ = 0.1, *t*_e_ is the time when *C*_outlet_/*C*_inlet_ is equal to 0.9, and *L_MTZ_* is the length of mass transfer zone that is calculated according to previous literature [43]. The corresponding characteristic parameters of the breakthrough curves under different operating conditions are summarized in Table 6.

As compared in Figure 8a, β-CD exhibits a low HCHO adsorption capacity (*q*) with a short adsorption saturation time (*t_e_*) of 1.4 h. Chitosan shows a high HCHO adsorption capacity and a long adsorption saturation time of 37.2 h. Both the HCHO adsorption capacity and the saturation time of CGC are extremely larger than the two reference samples. The HCHO adsorption on CGC is not saturated in 100 h of adsorption. Calculated by Eq. (1), the HCHO adsorption capacities are 0.2 mg/g for β-CD, 6.0 mg/g for chitosan, and 10.9 mg/g for CGC, respectively. The HCHO adsorption performance difference is related to the structure and surface properties among the adsorbents. Firstly, CGC (10.8 m^2^/g) has larger surface area than those of β-CD (0.4 m^2^/g) and chitosan (1.7 m^2^/g), which is beneficial for the adsorption process. In addition, the synergistic effects between β-CD and the functional group of chitosan also played a crucial role for the adsorption of HCHO. Similar results were observed for the adsorption of benzoic acid from wastewater on CGC [24]. According to the above results, CGC is very favorable for the improvement of the HCHO adsorption capacity of HCHO. As seen in Table 6, *q* and *t_b_* decrease significantly with increasing temperature, indicating that the adsorption of HCHO onto CGC is significantly affected by the temperature, which is attributed to a consequence of the exothermic reaction and weak adsorption interactions. The HCHO adsorption capacity over CGC gradually decreases from 10.9 mg/g to 1.2 mg/g, when the flow rate increases from around 28 mL/min with the influent concentration of 41.5 mg/m^3^ to 84 mL/min, with the influent concentration of 37.2 mg/m^3^, which is attributed to the decrease of contact time and increase of mass transfer zone *L*_MTZ_.

#### 3.2.3. Desorption and Reusability Study

The reusability of CGC was investigated by measuring the adsorption capacities of HCHO and their desorption properties, as displayed in Figure 9. The effect of temperature on HCHO desorption from CGC after saturated adsorption is shown in Figure 9a. HCHO can be completely desorbed from CGC at 60 °C with a desorption time of around 5 h. While the complete release of HCHO from CGC requires about 24 and 36 h at 40 and 20 °C, respectively. The result is consistent with the effect of temperature on HCHO adsorption discussed above. The result also implies that CGC can be easily regenerated at a high temperature. An adsorption-desorption experiment on CGC for four cycles at a flow rate of 28 mL/min and influent HCHO concentration of 41.5 mg/m^3^ was explored. As compared in Figure 9b, only a very slight drop in the adsorption capacity is observed after four cycles. These results suggest the good HCHO adsorption reusability of CGC, being fully satisfactory for practical applications.

In situ DRIFTS was used to study the HCHO adsorption of CGC with respect to the adsorbed specie on the adsorbent surface. Figure 10 shows the dynamic changes in the DRIFTS spectra of CGC as a function of time in a flow of gas + HCHO at 20 °C. After exposing the adsorbent to HCHO mixture gas, three bands appear at 1670, 1650, and 1020 cm^−1^. The peak at 1670 cm^−1^ corresponds to stretching vibrations of C=O of aldehyde group and its intensity increases with exposing time, indicating that HCHO molecules are adsorbed onto CGC. The peak at 1650 cm^−1^ is the characteristic stretching vibration of C=N, which is from the Schiff base reaction between the aldehyde group of HCHO and the amino group of chitosan. The peak at 1330 cm^−1^ due to stretching vibrations of C–N bond is also observed. The bands of –OH/N–H of chitosan and β-CD and aldehyde group of HCHO shift from 3400 to 3450 cm^−1^ and 1656 to 1670 cm^−1^, respectively, which is related to the hydrogen bond interactions between chitosan/β-CD and HCHO. The results indicate that the adsorption of HCHO over CGC can be significantly involved by the synergistic effects of the Schiff base reaction and hydrogen bond interaction, and the corresponding HCHO adsorption mechanism is proposed, as shown in Scheme 2.

## 4. Conclusions

Herein, we reported a kind of fully biodegradable materials, namely β-CD modified chitosan, or CGC, for the removal of indoor HCHO pollution with an enhanced performance. It was found that the abundant amino and hydroxyl groups in CGC played an important role for the adsorption of HCHO. The HCHO adsorption process by CGC was well described by the Freundlich isotherm and pseudo-second-order kinetic models. Further investigation of the adsorption mechanism revealed that the HCHO adsorption is a physicochemical adsorption with multi-molecular layer adsorption. Synergistic effects of the Schiff base reaction and the hydrogen bond interaction toward HCHO during the HCHO adsorption process was proposed. In short, we provided an effective design strategy and HCHO adsorption material.

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
