# Peer review of "Enhanced Formaldehyde Removal from Air Using Fully Biodegradable Chitosan Grafted β-Cyclodextrin Adsorbent with Weak Chemical Interaction"

_polymers, 2019, doi:10.3390/polym11020276_

Round 1
Reviewer 1 Report
After reading the manuscript entitled:
"Enhanced Formaldehyde Removal from Air using Fully Biodegradable Chitosan Grafted β-Cyclodextrin Adsorbent with Weak Chemical Interaction" I regret to say it is not suitable for publication in the current shape according to the following reasons: 1- When performing plagiarism check using Ithenticate software a 54% was obtained with references and 48% without. This is a very high percentage and it should not exceed 10%. 2- The following comments need to be addressed as will before considering it for publication: - In page 3, line 118 it was mentioned that a particle size analysis was performed for the adsorbent while in the discussion of results this analysis was not discussed. - In page 3, line 126 it was mentioned that Nitrogen adsorption was performed for the three samples and the results was tabulated in supplemantary data, while as an important data in should be in the body of the manuscript, furthermore, the isotherms of the three samples should be shown. - In page 6, line 201 it was mentioned that 60% porosity was obtained from SEM image in Figure 3, while this information is obtained from BET analysis so please check. -For figure 6, the effect of temperature on time for reaching maximum adsorption should be discussed in more depth taking into acount the effect of temperature on both rate of bulk and porous diffusion. -In Figure 6, it was mentioned in page 9, line 292 that the maximum adsorption capacity for 20C was achieved in 160h while the figure show about 105h was the saturation time. -Figure 7c need to be discussed deeper, it is clear that adsorption took place at two different stages, external and internal adsorption, and that internal adsorption is more significant than the external one. - In page 14, lines 396-398 are repetetion of the lines 394-396.Author Response
Comments from reviewer 1
"Enhanced Formaldehyde Removal from Air using Fully Biodegradable Chitosan Grafted β-Cyclodextrin Adsorbent with Weak Chemical Interaction".I regret to say it is not suitable for publication in the current shape according to the following reasons:
Answer: Thank you so much for your valuable comments and suggestions. We would like to thank the editor for giving us a chance to resubmit the paper, and also thank the reviewers for giving us some suggestions, which would help us both in English and in depth to improve the quality of the paper. Here we submit a new version of our manuscript, which has been modified according to the reviewers’ suggestions. Efforts were also made to correct the mistakes and improve the English of the manuscript.
Question1:When performing plagiarism check using Ithenticate software a 54% was obtained with references and 48% without. This is a very high percentage and it should not exceed 10%.
Answer: Thank you so much for your valuable comments and suggestions. We have read our manuscript very carefully for English language accuracy and we have corrected it in the revised version. We believe that it will meet the required standard of English.
Question2:The following comments need to be addressed as will before considering it for publication:In page 3, line 118 it was mentioned that a particle size analysis was performed for the adsorbent while in the discussion of results this analysis was not discussed.
Answer: Thanks for the comments. We have discussed it in the revised version (Page 6 line 118).
Question3: In page 3, line 126 it was mentioned that Nitrogen adsorption was performed for the three samples and the results was tabulated in supplemantary data, while as an important data in should be in the body of the manuscript, furthermore, the isotherms of the three samples should be shown.
Answer: Thanks for the comments. We have added them in the revised manuscript(Page 5 line 118).
Question4:In page 6, line 201 it was mentioned that 60% porosity was obtained from SEM image in Figure 3, while this information is obtained from BET analysis so please check.
Answer: Thanks for the comments. We have corrected it in the revised manuscript(Page 6 line 165-166).
Question5:For figure 6, the effect of temperature on time for reaching maximum adsorption should be discussed in more depth taking into acount the effect of temperature on both rate of bulk and porous diffusion.
Answer: Thanks for the comments. We have discussed it in the revised manuscript(Page 8 line 246-249).
Question6:In Figure 6, it was mentioned in page 9, line 292 that the maximum adsorption capacity for 20C was achieved in 160h while the figure show about 105h was the saturation time.
Answer: Thanks for the comments. We have corrected it in the revised manuscript(Page 8 line 245).
Question7:Figure 7c need to be discussed deeper, it is clear that adsorption took place at two different stages, external and internal adsorption, and that internal adsorption is more significant than the external one. - In page 14, lines 396-398 are repetetion of the lines 394-396.
Answer: Thanks for the comments. We have discussed it in the revised manuscript(Page 9 line 285-292).

Reviewer 2 Report
In this manuscript, the authors described an application of chitosan grafted β-cyclodextrin (CGC) to adsorbent of formaldehyde. The authors prepared CGC according to the method reported previously, characterized the CGC sample by IR, TG/DTA, SEM, and powder XRD, and studied the equilibrium and kinetics of formaldehyde adsorption. The adsorption data have been analyzed using several models. On the basis of the data indicated, the authors have demonstrated the advantage of CGC as a formaldehyde adsorbent. I would recommend publication of this paper in Polymers after revision.
The points to be attended to are listed below.
(1) The degree of modification of β-CD (the β-CD content) should be determined.
(2) The authors should discuss why the grafting of β-CD increases the porosity. This is because the porosity is an important point for adsorbents. Is it possible to prepare CGC samples of different porosities, possessing the same β-CD content?
(3) It may be better to compare the porosity and adsorption properties of CGC samples with different β-CD contents.
(4) TG/DTA data for β-CD and chitosan should be added to Figure 2.
(5) The authors could indicate the theoretical curves based on the D-R model in Figure 5.
(6) The authors should discuss why the adsorption isotherms and kinetics obey the Freundlich model and a pseudo-second-order model, respectively, and describe something about the values of parameters.
Author Response
Comments from reviewer 2
In this manuscript, the authors described an application of chitosan grafted β-cyclodextrin (CGC) to adsorbent of formaldehyde. The authors prepared CGC according to the method reported previously, characterized the CGC sample by IR, TG/DTA, SEM, and powder XRD, and studied the equilibrium and kinetics of formaldehyde adsorption. The adsorption data have been analyzed using several models. On the basis of the data indicated, the authors have demonstrated the advantage of CGC as a formaldehyde adsorbent. I would recommend publication of this paper in Polymers after revision.
Answer: Thanks for the comments. We have read our manuscript very carefully and we have corrected it in the revised version. We believe that it will meet the required standard of English.
Question 1: The degree of modification of β-CD (the β-CD content) should be determined.
Answer: Thank you for the suggestion. As shown in Table 1(Page 3 line 128), β-CD content in CGC was 38.62 μmol/g.
Question 2: The authors should discuss why the grafting of β-CD increases the porosity. This is because the porosity is an important point for adsorbents. Is it possible to prepare CGC samples of different porosities, possessing the same β-CD content?
Answer: Thank you. Figure 3a also indicated that β-CD has a dense crystal structure, and after immobilization onto chitosan, CGC has an ordered porous structure with a pore size range of 5–20 nm, indicating that it is beneficial for the diffusion and adsorption of the pollutant gas. For the same β-CD content, it Is possible to prepare CGC samples of different porosities according to different preparation process.
Question 3: It may be better to compare the porosity and adsorption properties of CGC samples with different β-CD contents.
Answer: Thank you for the suggestion. It is important for HCHO adsorption with the different porosity. In the previously published ref, we have compared the similar experiment, and the results showed microstructure with some porosity in the CGC has better adsorption performance.
Question 4: TG/DTA data for β-CD and chitosan should be added to Figure 2.
Answer: Thanks for the comments. We have added it in the revised manuscript.
Question 5: The authors could indicate the theoretical curves based on the D-R model in Figure 5.
Answer: Thanks for the comments. We have added it in the revised manuscript.
Question 5:The authors should discuss why the adsorption isotherms and kinetics obey the Freundlich model and a pseudo-second-order model, respectively, and describe something about the values of parameters.
Answer: Thanks for the comments. The best fitted model was then selected on the basis of the correlation coefficient (R2) of the linear regression of the experimental data with the proposed models. The values of parameters were listed in Table 2 and Table 4 in the revised manuscript, respectively.

Round 2
Reviewer 1 Report
Sorry to say that plagiarism ratio still very high, 38% without a bibliography and 48% with bibliography.
Author Response
Question1: Sorry to say that plagiarism ratio still very high, 38% without a bibliography and 48% with bibliography.
Answer: Thank you so much for your valuable comments and suggestions. We have revised the version, and the rest duplication is mainly from the reference list. Some refs have been deleted, and we hope that it will meet the required standard of English.

Reviewer 2 Report
The authors have responded and answered the questions and comments I raised previously. I would recommend publication of this manuscript in Polymers, although I think that the data in Tables 2, 4, 5, and 6 should indicate the errors (the standard deviations).
Author Response
Question 1: The authors have responded and answered the questions and comments I raised previously. I would recommend publication of this manuscript in Polymers, although I think that the data in Tables 2, 4, 5, and 6 should indicate the errors (the standard deviations).
Answer: Thanks for the comments. As shown in the Tables 2,4,5, and 6, all the errors were in the range of 0.5%-5%. We have consulted the relevant literatures, and generally we do not list the errors in the paper.
